# Knee Angle Affects Posterior Chain Muscle Activation During an Isometric Test Used in Soccer Players

**DOI:** 10.3390/sports7010013

**Published:** 2019-01-04

**Authors:** Paul James Read, Anthony Nicholas Turner, Richard Clarke, Samuel Applebee, Jonathan Hughes

**Affiliations:** 1Athlete Health and Performance Research Centre, Aspetar Orthopaedic and Sports Medicine Hospital, Doha P.O. Box 29222, Qatar; 2London Sports Institute, Middlesex University, London NW4 4BT, UK; a.n.turner@mdx.ac.uk; 3School of Sport and Exercise, University of Gloucestershire, Gloucester GL2 9HW, UK; rclarke@glos.ac.uk (R.C.); samuel.applebee@gmail.com (S.A.); jhughes1@glos.ac.uk (J.H.)

**Keywords:** hamstring, strength test, muscle activation

## Abstract

Background: It has been suggested that altering the knee flexion angle during a commonly used supine isometric strength test developed with professional soccer players changes preferential hamstring muscle recruitment. The aim of this study was to examine the electromyography (EMG) knee joint-angle relationship during this test, as these data are currently unknown. Methods: Ten recreational male soccer athletes (age: 28 ± 2.4 years) were recruited and performed a supine isometric strength test on their dominant leg with the knee placed at two pre-selected flexion angles (30° and 90°). The surface EMG of the gluteus maximus, biceps femoris, semitendinosus, and medial gastrocnemius was measured, in addition to the within-session reliability (intraclass correlation coefficient (ICC) and coefficient of variation (CV)). Results: Within-session reliability showed large variation dependent upon the test position and muscle measured (CV% = 8.8–36.1) Absolute mean EMG activity and percentage of maximum voluntary isometric contraction (MVIC) indicated different magnitudes of activation between the two test positions; however, significant mean differences were present for the biceps femoris only with greater activation recorded at the 30° knee angle (% MVIC: 31 ± 9 vs. 22 ± 7; *p* = 0.002). These differences (30% mean difference) were greater than the observed typical measurement error (CV% = 13.1–14.3 for the 90° and 30° test positions, respectively). Furthermore, the percentage MVIC showed a trend of heightened activation of all muscles with the knee positioned at 30°, but there was also more within-subject variation, and this was more pronounced for the gluteus maximus (CV% = 36.1 vs. 19.8) and medial gastrocnemius (CV% 31 vs. 22.6). Conclusions: These results indicate that biceps femoris and overall posterior chain muscle activation is increased with the knee positioned at 30° of flexion; however, the 90° angle displayed less variation in performance within individual participants, especially in the gluteus maximus and medial gastrocnemius. Thus, practitioners using this test to assess hamstring muscle strength should ensure appropriate familiarisation is afforded, and then may wish to prioritise the 30° knee position.

## 1. Introduction

Participation in competitive soccer training and match-play places significant physical and physiological demands on athletes [1], resulting in an inherent risk of injury. Hamstring strains are the most common injury in elite adult soccer [2,3] and also frequently occur in elite male youth soccer players [4,5]. The knee flexors are also key antagonist muscles that contribute to knee joint stabilisation, reducing the magnitude of anterior shear force during high velocity actions [6]. Previous data indicate that hamstring strength [7] and between-limb imbalances are risk factors for injury [8]. Strength assessment is thus warranted in order to identify soccer players who may be at a heightened injury risk, allowing more individually targeted training interventions.

Assessment of hamstring strength has traditionally been conducted using isokinetic dynamometry [9,10,11]. These tests have limited utility as a screening tool to identify players who subsequently sustain a hamstring injury [12] and are not practically viable for screening large numbers of athletes because of time inefficiency and the need for expensive laboratory equipment. An alternative approach is to examine hamstring strength using isometric protocols that involve minimal structural muscle damage [13] and can thus be used regularly in-season. Additionally, rate of force production can more easily be quantified [14], which has connotations for soccer players where constraints exist for the time availability of force production.

A simple and practically viable test has recently been developed in professional soccer players to measure isometric hamstring strength and asymmetry using force plate diagnostics [14]. This assessment is reliable (intraclass correlation coefficient (ICC) = 0.93–0.95; coefficient of variation (CV) = 4.3–6.3%) and has shown sensitivity to identify changes in performance following soccer match play [14]. Field-based measurements that examine hamstring strength and monitor both acute and chronic changes following soccer training and competitions can be readily used in applied settings. However, before this approach can be recommended to coaches, it is necessary to examine the level of muscle activation during the execution of the task. This information is required to ensure time efficiency in testing and that the correct muscles are targeted, which has implications for injury risk screening, longitudinal monitoring, and training prescription [15]. 

The research by McCall et al. [14] included two different test positions; specifically, lying supine, with the foot placed on top of a box and the knee positioned at both 30° and 90° of flexion. In the development of this assessment, it was suggested that altering the knee position changes the recruitment of the hamstring muscles [14] based on previous data [16]. The lateral and medial hamstrings have been shown to be maximally activated at 15° to 30° and 90° to 105° of knee flexion respectively [16]; however, these values were obtained during an open chain isokinetic test. Onishi et al. [16] also included isometric tests that showed greater activation of the biceps femoris at lower knee flexion angles, but it should be acknowledged that this test was performed in a prone position with the knee placed at both 60° and 90° of flexion. In addition, several other muscles will contribute to the development of peak force force during this test, such as the gluteals and plantar flexors; their level of activation is currently unknown. 

The aim of this study was to examine the level of muscle activation in the biceps femoris, medial hamstrings, gluteus maximus, and gastrocnemius during a previously used isometric hamstring strength test [14] with the knee at two designated positions, as previously proposed (either 30° and 90° of flexion). It was hypothesised that hamstring recruitment and, in particular, biceps femoris activation would be greater at the shallower knee angle (30°) as a result of magnified muscle activity when the bi-articular hamstrings are at peak stretch [17,18].

## 2. Materials and Methods

### 2.1. Participants

Ten recreational male soccer players (age: 28 ± 2.4 year; height: 177 ± 25 cm; body mass: 80.5 ± 6.4 kg) volunteered for this study. Inclusion criteria were as follows: (1) to have undertaken a minimum of six months of resistance training encompassing lower limb and posterior chain exercises; (2) participating regularly in soccer training and competitions; (3) free from injury within the last three months; and (4) no previous history of hamstring strain injury. Physical activity readiness questionnaires were collected prior to the commencement of testing. Ethical approval (JHUGHES17-18) was granted by the institutional ethics committee in accordance with the declaration of Helsinki. Participants provided informed consent and were reserved the right to withdraw from the study at any time without consequence. 

### 2.2. Experimental Design

A cross-sectional design was used to examine muscle activation during a recently developed field-based isometric hamstring strength test. Participants were required to attend the laboratory on two separate occasions separated by a period of seven days. The first session was used to familiarise participants with the assessment protocols and equipment, and the second session was used for data collection. Standardised procedures were replicated at each test session including the warm up, test set-up, and participant instructions. Participants were asked to refrain from strenuous exercise for at least 48 h prior to testing and to eat according to their normal diet, and drink only water one hour prior to each test session. 

### 2.3. Procedures

The test was performed on the dominant leg only with the knee placed in flexion at two pre-selected angles (30° and 90°; Figure 1a,b where the 30° position corresponds to an internal joint angle of 150°) using a manual goniometer, as previously suggested [14]. Participants were positioned supine and asked to lie flat on an exercise mat. The heel of the dominant limb was then placed on top of a 30 cm box, while the non-working limb lay flat on the floor facing forwards. Standardised encouragement was provided, with instructions to push their heel as hard as possible into the box and hold the contraction for three seconds while simultaneously keeping their head, upper body, hands, and buttocks flat on the floor [14]. To restrict vertical elevation of the contralateral hip, a member of the research team provided manual resistance to the participant throughout the test, ensuring maintenance of the correct position [14]. The order of testing for each respective knee angle was counterbalanced to avoid an order effect. Three repetitions were performed on the dominant leg only with two minutes of recovery between trials based on previous recommendations [14]. 

### 2.4. Surface Electromyography (sEMG)

The subjects’ body hair was shaved at the site of electrode placement, and the skin was cleaned with an alcohol wipe and lightly abrased to remove dead skin cells before affixing the surface EMG electrode. Bipolar active disposable dual Ag/AgCl snap electrodes spanning 1 cm in diameter for each circular conductive area with 2 cm center-to-center spacing were used for all trials. Electrodes were placed on the dominant limb along the axes of the muscle fibers for the gluteus maximus (GM), medial hamstrings (MH), biceps femoris (BF), and medial gastrocnemius (MG) according to the Surface Electromyography for the Non-Invasive Assessment of Muscles (SENIAM) protocol [19]. A ground electrode was placed on the right medial malleolus. The sEMG signals were recorded by an EMG acquisition system (Biometrics MWX8 DataLog; Biometrics Ltd., Newport, UK) with a sampling rate of 1000 Hz using a commercially designed software program (Biometrics DATALOG Software v8.51: Biometrics Ltd., Newport, UK). EMG activity was amplified (bipolar differential amplifier, input impedance = 2 MΩ, common mode rejection ratio 100 dB minute [60 Hz], gain x 20, noise > 5 µV) and converted from an analog to digital signal (12 bit). The digitised surface EMG data were band-pass filtered at 20–400 Hz using a fourth-order Butterworth filter with a zero lag. For muscle activation time domain analysis, root mean square (RMS) (30 ms moving window) was calculated during the maximum voluntary isometric contraction (MVIC). The sEMG data were then normalised to the RMS average of the peak MVICs for each amplitude and muscle. The RMS analysis was defined from the average of the first three repetitions for each condition and muscle.

EMG signals collected during all conditions were normalised to a maximum voluntary isometric contraction (MVIC) against a fixed resistance. Two trials of 5 s MVICs were performed for each muscle with a 1 min rest interval between actions for the dominant leg. The first MVIC was performed to familiarise the participant with the procedure. For GM, BF, and MH MVIC, subjects were in the prone position with their knee fully extended and locked into a leg curl bench (Power Lift, Jefferson, IA, USA) with the restraint placed on the distal region of the Achilles tendon as the subject attempted to maximally and simultaneously extend the hip and flex the knee. For MG MVICs, subjects stood with their knee fully extended and with vertical resistance through the shoulders via fixed squat rack while they attempted to maximally plantar flex. Standardised, verbal encouragement was given during all MVICs. The order of MVICs was counterbalanced to reduce the potential for neuromuscular fatigue.

### 2.5. Statistical Analysis 

Descriptive statistics (mean ± SD) for both mean and peak EMG were calculated for each test position. A Shapiro–Wilks test was used to check the normality distribution of the data. To examine potential differences in the level of muscle activation between test positions, a two-way repeated measure analysis of variance (ANOVA) with two factors (knee angle (two levels); and muscle (four levels)) was utilised with statistical significance set at *p* < 0.05. Cohen’s *d* effect sizes were also calculated to interpret the magnitude of change in the RMS EMG signal for each muscle measured between test positions using the following classifications: standardised mean differences of 0.2, 0.5, and 0.8 for small, medium, and large effect sizes, respectively [20]. Within-session reliability was assessed for each muscle measured in both the 30° and 90° test positions via intraclass correlation coefficient (ICC) using a two-way random model with absolute agreement and the coefficient of variation (CV = (SD/mean × 100). The latter analysis was included to allow further interpretation of the magnitude of change in muscle activation between test positions.

## 3. Results

Within-session reliability statistics are shown in Table 1. Absolute mean EMG activity indicated different magnitudes of hamstring, gluteal, and plantar flexor activation between the two test positions; however, significant mean differences were present only for the biceps femoris, where greater absolute values were shown at the 30° knee angle *F*_(1–9)_ = 19.207 (*p* = 0.002; *d* = 1.19). EMG percentage MVIC and percentage differences between the two knee positions (Table 2) showed a trend of greater muscle activation with the knee positioned at 30°. Significant differences were shown between test positions for the biceps femoris only corresponding to a very large effect size (*p* < 0.05; *d* = 1.19). Medial hamstring, gastrocnemius, and gluteus maximus activation was higher with the knee positioned at 30°, but these differences were not meaningful and effect sizes were small (*d* = 0.07–0.40).

## 4. Discussion

The aim of this study was to examine the level of posterior chain muscle activation during a hamstring strength test developed for soccer players. It has previously been suggested that a change in knee angle is required to preferentially alter muscle recruitment; specifically, activation of the medial hamstrings and biceps femoris [14]. The results showed that while changes in muscle activation of the medial hamstrings and biceps femoris were present in the two test positions (30° and 90° of knee flexion) with heightened activation observed with the knee positioned at 30°, significant differences were observed for the biceps femoris only, which confirms the test hypothesis.

Greater biceps femoris, medial hamstring, and gluteal activation were seen in the current study with the knee positioned at 30° of flexion. This is likely the result of magnified muscle activation when the hamstrings are in an elongated position [17,18] and increases in the distal moment arm [21]. Also, peak torque of the knee flexors has been shown to occur with knee angles between 21–49° of knee flexion, with reductions in torque concomitant with greater flexion angles [16]. Thus, greater recruitment of key posterior chain musculature can be expected with the knee positioned at 30° of flexion during this supine isometric closed chain strength assessment.

It has previously been suggested [14] that biceps femoris and medial hamstring activation peaks between 15–30° and 90–105° of knee flexion, respectively; however, these results were obtained during an open chain, isokinetic test [16]. Previous research has shown differences in the magnitude of muscle activity and preferential muscle recruitment between open and closed chain exercises [22]. The original recommendations in the development of the novel isometric posterior chain lower limb muscle test examined in the current study were to alter the knee flexion position to preferentially recruit either the biceps femoris (30°) or medial hamstrings (90°) [14]. The results of the current study do not support this notion as greater recruitment of both the biceps femoris and medial hamstrings were observed with the knee positioned at 30° of flexion. Previous literature has shown no changes in EMG activity with different flexion angles, albeit at the elbow joint [23]. Conversely, Lunnen et al. [21] observed changes in biceps femoris activation by altering the angle of hip flexion, which was consistent with the current study; however, it should be noted that the test set up used by Lunnen et al. [21] was different to the present study because in their work, the knee position was maintained at 60°. The assessment proposed by McCall et al. [14] is performed in supine and requires a closed kinetic chain test position, allowing greater hip extension force production in comparison with open chain tasks and higher biceps femoris long head activation [24].

When interpreting the results of this study, it should be considered that the ICC values were excellent, indicating good between-participant variance [25] and maintenance of rank order between trials. However, greater variations in muscle activation were present, as indicated by large standard deviations for the medial hamstrings, gastrocnemius, and gluteus maximum. In addition, the CV% shown in the medial gastrocnemius and gluteus maximus would be considered as being large [26], with lower values seen in the medial hamstrings and biceps femoris (Table 1). These data indicate that pronounced variation in performance was shown within individual participants. This was consistent across both test positions and may be in part because of the participants’ interpretation of the task. In spite of this, the CV values reported were lower than the percentage differences between test positions for the medial hamstrings and biceps femoris (Table 2), suggesting that the observed differences were ‘real’ and indicative of a meaningful change. 

The observed within-subject variability in the supine test position could be in part because of the instructions to ‘push’ with their heel into the box [14], which may create confusion in how the participant should produce their maximum force. The term ‘push’ may be associated with an attentional focus of moving an object away from their body, encouraging a hip extension dominant production of force in some participants. This interpretation is supported by the need of the researcher to anchor the hips of the participant to the floor [14]. Conversely, some participants may have produced force by driving the heel into the box and towards their hip via a greater contribution of knee flexion. Cumulatively, this may have contributed to the greater variation in either gluteal and gastrocnemius contributions during both test positions. It should, therefore, be considered that the instruction to ‘push’ may not be optimal in order to standardise muscle recruitment strategies, and alternate cues warrant further investigation. The level of medial gastrocnemius activation could also be affected by the position of the ankle where the athlete may have plantar flexed or dorsiflexed to a greater amount. Thus, practitioners are advised to provide adequate standardisation and familiarisation prior to testing and ensure that participants are clear on the task requirements to obtain reliable measures that can be used for test/re-test comparison following targeted training interventions.

## 5. Conclusions

The current study indicates that altering the knee position to 90° of flexion does not preferentially recruit the medial hamstrings, as has been previously suggested. Significantly greater biceps femoris activation was present with a shallower knee flexion angle, which was also combined with heightened recruitment of other key posterior chain musculature, including the medial hamstrings and gluteus maximus. However, caution should be applied as these differences were not statistically significant and pronounced within-subject variability was present. On the basis of these results, it could be recommended that following a period of extensive familiarisation, practitioners wishing to use this test as part of a regular screening and monitoring programme with soccer players should prioritise the 30° knee flexion position because of the observed increases in posterior chain muscle recruitment. Deficiencies measured during this test could be used to identify baseline injury risk factors for hamstring strains and knee ligamentous injury, in addition to monitoring the effectiveness of targeted strength and conditioning programs.

## Figures and Tables

**Figure 1 sports-07-00013-f001:**
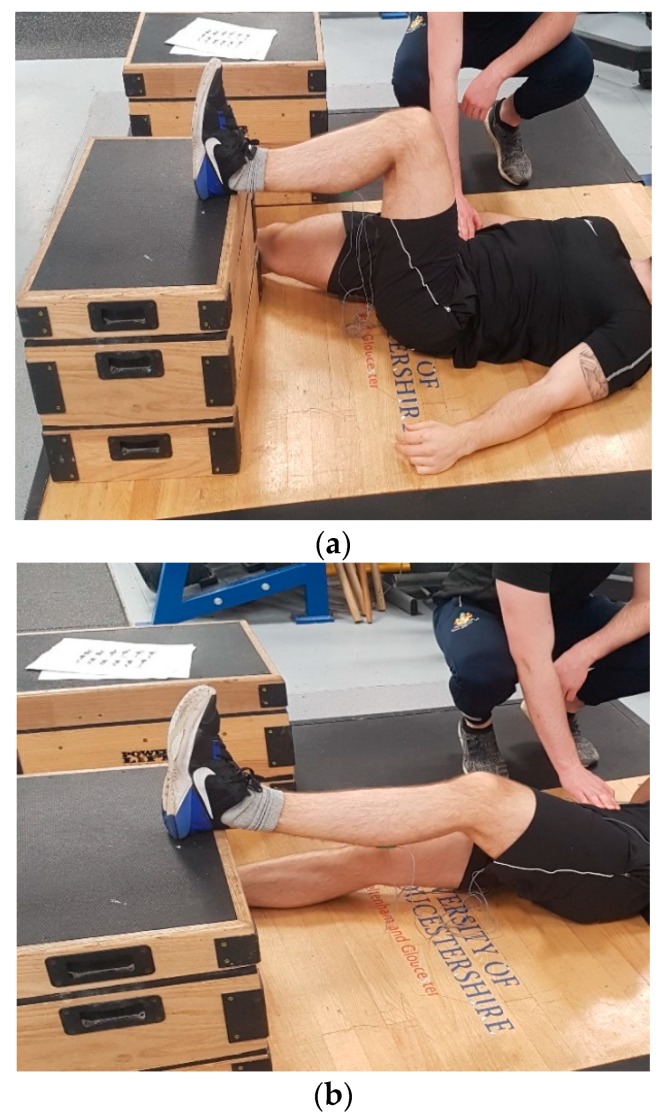
Test position of the isometric posterior chain lower limb muscle test measured at 90° (**a**) and 30° (**b**) of knee flexion.

**Table 1 sports-07-00013-t001:** Reliability statistics for each muscle measured in both test positions. ICC—intraclass correlation coefficient; CV—coefficient of variation.

	30° Knee Angle		90° Knee Angle	
	ICC	CV%	ICC	CV%
Medial gastrocnemius	0.93	31%	0.97	22.6%
Medial hamstrings	0.96	13.1%	0.88	8.8%
Biceps femoris	0.95	14.3%	0.95	13.1%
Gluteus maximum	0.96	36.1%	0.99	19.8%

**Table 2 sports-07-00013-t002:** Percentage maximum voluntary isometric contraction (MVIC) for each muscle at the two different knee flexion angles.

	Knee Flexion Angle (Degrees)			
% MVIC	30°	90°	% Difference	Effect Size *d*
Medial gastrocnemius	15 ± 11	14 ± 10	5	0.07
Medial hamstrings	34 ± 14	30 ± 12	15	0.40
Biceps femoris	31 ± 9	22 ± 7	30	1.19
Gluteus maximus	27 ± 10	24 ± 11	7	0.18

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
