# Peer review of "Knee Angle Affects Posterior Chain Muscle Activation During an Isometric Test Used in Soccer Players"

_sports, 2019, doi:10.3390/sports7010013_

Round 1

Reviewer 1 Report

Overall

The authors have presented an interesting study that compares the magnitude and proportional contribution of selected lower-limb extensor muscles. The findings of this investigation have implications for testing, monitoring, and injury screening.

I was confused somewhat by the term ‘percentage contribution’. Do the authors mean % of MVC?

Other concerns are the cues given, the reliability of the measures and some grammatical issues.

Please see below for my specific comments.   

Line 15-16 This sentence reads awkwardly. Consider reworking.

Line 25-26. This statement is unclear. Do the authors mean %MVC or proportional contribution with respect to the other measured muscles?

Line 34 The structure of this sentence can be improved. Consider something like "... resulting in an inherent risk of injury"

Line 45-48 This is a run-on sentence and needs to be reworked. You also need references to support the second statement in this sentence - that there is evidence that that such actions are unsuitable for the population you describe. If not, please remove this statement. 

Line 51-52 This is a good reason to justify isometric testing. Please provide references to support this statement though.

Line 58-60 Please rework this sentence. Also, I think that you can find a stronger justification for measuring muscle activation. Why is it necessary to better understand the muscle activation patterns?

Line 77 You're not measuring the entire hamstring musculature. What you're referring to is certain hamstring muscles of interest.

Line 87-89 Please confirm that informed consent was acquired.

Line 92 ‘attend’ Attend what?

Line 100-101 Knee joint angle: Please confirm that in this case it was internal knee angle, and full extension represented 180 degrees. If not, please clarify.

Line 104 Was there any cue regarding the rate of effort (i.e. hard and fast)? A hard push can be either a ramped rise over 2-3 second to a peak, which is then held, or it can be max RFD with the objective to reach peak force as soon as possible. These are considered different qualities, so please be specific.

Line 111 I understand that reliability data has previously been published, but can you provide reliability data from this cohort and your lab?

Line 113 Was the skin also lightly abrased to remove dead skin cells?

Line 125 Can you justify why you chose this filter?

Line 128 Please provide reliability data for your EMG measures.

Line 146. I think you need to be careful about how you word this, as it is currently confusing. Do you mean percent of MVC for each measured muscle?

Line 166 This sentence is awkward and needs reworking.

Line 168-170 Do not restate your findings in the discussion.

Line 178-180 Explain why would you expect measures from an isokinetic, open chain action to differ from the action you used.

Line 199-204 This highlights why reliability data is needed.

Author Response

Reviewer 1 Overall Comments.

The authors have presented an interesting study that compares the magnitude and proportional contribution of selected lower-limb extensor muscles. The findings of this investigation have implications for testing, monitoring, and injury screening. I was confused somewhat by the term ‘percentage contribution’. Do the authors mean % of MVC? Other concerns are the cues given, the reliability of the measures and some grammatical issues.

Author response: Thank you for taking the time to review our work. Your expertise is highly valued and we have made changes in accordance with your comments which we believe have enhanced the quality of the manuscript and these have been indicated below:

Specific comments.   

Line 15-16 This sentence reads awkwardly. Consider reworking.

Author response: We have simplified this sentence to improve readability.

Line 25-26. This statement is unclear. Do the authors mean %MVC or proportional contribution with respect to the other measured muscles?

Author response: Thank you for comment here and allowing us to clarify. We are referring to the %MVC and the text in this section has been amended accordingly.

Line 34 The structure of this sentence can be improved. Consider something like "... resulting in an inherent risk of injury"

Author response: This sentence has been amended in-line with your suggestion.

Line 45-48 This is a run-on sentence and needs to be reworked. You also need references to support the second statement in this sentence - that there is evidence that that such actions are unsuitable for the population you describe. If not, please remove this statement. 

Author response: We agree the sentence is a run-on and have therefore removed it from the manuscript.

Line 51-52 This is a good reason to justify isometric testing. Please provide references to support this statement though.

Author response: We agree this is an important point that requires further supporting evidence and have now included the following citation: Maffiuletti et al. Rate of force development: physiological and methodological considerations. Eur J Appl Physiol, 2016.

Line 58-60 Please rework this sentence. Also, I think that you can find a stronger justification for measuring muscle activation. Why is it necessary to better understand the muscle activation patterns?

Author response: This sentence has been edited and further information has been provided to more clearly outline why understanding the muscle activation during this test is needed

Line 77 You're not measuring the entire hamstring musculature. What you're referring to is certain hamstring muscles of interest.

Author response: The word ‘total’ has been removed to more accurately reflect what is being measured during this study.

Line 87-89 Please confirm that informed consent was acquired.

Author response: Text has been added at the end of the participants section to state that informed consent was provided prior to participation.

Line 92 ‘attend’ Attend what?

Author response: Text has been added to state that participants attended our laboratory.

Line 100-101 Knee joint angle: Please confirm that in this case it was internal knee angle, and full extension represented 180 degrees. If not, please clarify.

Author response: The knee angle is degrees of flexion. Thus, we can confirm, 180 degrees would represent full extension (with 0° of knee flexion).

Line 104 Was there any cue regarding the rate of effort (i.e. hard and fast)? A hard push can be either a ramped rise over 2-3 second to a peak, which is then held, or it can be max RFD with the objective to reach peak force as soon as possible. These are considered different qualities, so please be specific.

Author response: The cue was to push ‘as hard as possible’ to ensure replication of the procedures stated by McCall et al. (2015) to comply with the aims and purpose of the study.

Line 111 I understand that reliability data has previously been published, but can you provide reliability data from this cohort and your lab?

Author response: For the purpose of this study, force data were not collected as the aim was to examine the muscle activity at the two respective knee angles. Thus, reliability of force metrics was not examine.

Line 113 Was the skin also lightly abrased to remove dead skin cells?

Author response: Thank you for your comment. We can confirm the skin was abrased to remove dead skin cells and we have now added this to the manuscript.

Line 125 Can you justify why you chose this filter?

Author response: We used the low frequency band pass filter to remove baseline drift that can sometimes be associated with additional movements during testing procedures. The usual rates of low frequency filtering are between are 5 to 20 Hz. The use of the high frequency band pass filter is to remove high frequency noise not associated with the movement being measured. It is recommended by SENIAM that the high frequency filter should be high enough to allow for sudden on-off bursts of the EMG to be visible and accounted for. The usual rates for high filters are 200 Hz – 1 kHz. We went with the SENIAM recommendations for surface EMG: 20 Hz -  500 Hz filter.

If you feel the readers require this level of detail we are more than happy to add the text into the manuscript.

Line 128 Please provide reliability data for your EMG measures.

Author response: Reliability data has now been provided with references now made to this in the methods, results and discussion. Furthermore, these data have been used to inform the previously included discussion regarding variability in task execution related to cueing and also to determine more clearly if the observed changes seen between test positions were ‘real’ and indicative of a meaningful change.

Line 146. I think you need to be careful about how you word this, as it is currently confusing. Do you mean percent of MVC for each measured muscle?

Author response: Thank you for allowing us to clarify. The text has been amended to state change in %MVC for each muscle measured.

Line 166 This sentence is awkward and needs reworking.

Author response: The first two sentences of this paragraph have been edited and simplified in line with your suggestions.

Line 168-170 Do not restate your findings in the discussion.

Author response: Thank you for your comment. In the opening paragraph of the discussion we have attempted to succinctly re-state the the aim and provide for the readers the main finding of the study and how it relates to our initial hypothesis. Following this, we have interpreted the data using supporting evidence in the remainder of this section. We believe this to be consistent with common guidelines for manuscript preparation. Therefore, we would request that the manuscript remains un-altered here.

Line 178-180 Explain why would you expect measures from an isokinetic, open chain action to differ from the action you used.

Author response: Text has now been added to support this statement including literature which has shown differences in both the magnitude of muscle activity and preferential recruitment of selected muscles between open and closed chain exercises.

Line 199-204 This highlights why reliability data is needed.

Author response: These data have now been included in the manuscript as described in our previous response and the data have been used to further inform the discussion here.

Reviewer 2 Report

Hamstring strains are a common injury of elite soccer players. The study aims to clearify the knee angle effects during a field-based isometric posterior chain lower-limb muscle test recording SEMGs from the gluteus maximus, the biceps femoris, the semitendinosus and the medial gastrocnemius. In the two selected knee flexion angles of 30° and 90°, significantly greater activation was only found at the 30° knee angle for the biceps femoris with an overall trend of posterior chain muscle activation. The authors suggest therefore that practitioners should prioritise the 30° flexion when testing the contractile strength of hamstring muscle. The study carried out with ten male soccer players has been very well designed and is meticulously documented in the manuscript. The data are adequately statistically analyzed and clearly described and presented. The discussion is critical and constructive giving reference to other relevant studies. I suggest only very minor corrections:

l. 4 please add a question mark if your title asks a question ;-)

l. 24 insert the s for biceps

l. 159 Fig.1, the significance * is missing  in the diagram

l. 161 insert the s for biceps...

Author Response

Thank you for taking the time to review our manuscript. We appreciate your positive comments and expertise in this area. We have made the relevant changes in the manuscript as per your suggestions which are shown below.

l. 4 please add a question mark if your title asks a question ;-)

Author response: Question mark now included. Thank you for your observation.

l. 24 and l. 161 insert the s for biceps

Author response: Here and throughout the text has been corrected to read biceps femurs

l. 159 Fig.1, the significance * is missing in the diagram

Author response: Significance now included. Thank you for your observation.

Round 2

Reviewer 1 Report

Dear Authors,

Thank you for your thorough response to my comments. I have only a couple of remaining suggestions to consider.

It would be worth including an image (as a Figure) of the test itself. Because this paper centre’s on this assessment method, it would be of great value to the reader attempting to visualize the task. For this reason I have indicated ‘Can be improved’.

Figure 1 ‘% Muscle contributions’. This is better described as ‘Percent MVIC’. For this reason I have indicated ‘Can be improved’.

Please include a reference to support line 191-192 (‘It has previously been suggested…’)

Regards,

Lachlan James

Author Response

Reviewer comments

Thank you for your thorough response to my comments. I have only a couple of remaining suggestions to consider.

Author response: Thank you again for your time and expertise in reviewing our work and enhancing the quality of the manuscript. We have amended the manuscript in accordance with your comments below.

Specific comments:

It would be worth including an image (as a Figure) of the test itself. Because this paper centre’s on this assessment method, it would be of great value to the reader attempting to visualize the task.

Author response: Figures included as recommended

Figure 1 ‘% Muscle contributions’. This is better described as ‘Percent MVIC’. For this reason I have indicated ‘Can be improved’.

Author response: This has been amended as suggested

Please include a reference to support line 191-192 (‘It has previously been suggested…’)

Author response: Reference has been added as suggested to support the statement provided.